# Microencapsulation and Probiotic Characterization of *Lactiplantibacillus plantarum* LM-20: Therapeutic Application in a Murine Model of Ulcerative Colitis

**DOI:** 10.3390/nu17050749

**Published:** 2025-02-20

**Authors:** Cynthia Garfias Noguez, Morayma Ramírez Damián, Alicia Ortiz Moreno, Yazmín Karina Márquez Flores, Liliana Alamilla Beltrán, Mario Márquez Lemus, Luis G. Bermúdez Humarán, María Elena Sánchez Pardo

**Affiliations:** 1Escuela Nacional de Ciencias Biológicas, Instituto Politécnico Nacional, Campus Zacatenco, Unidad Profesional Adolfo López Mateos, Zacatenco, Av. Wilfrido Massieu 399, Colonia Nueva Industrial Vallejo, Alcaldía Gustavo A. Madero, Ciudad de México 07738, Mexico; cgarfez@gmail.com (C.G.N.); morayma.ramirez@outlook.com (M.R.D.); ortizalicia@hotmail.com (A.O.M.); ymarquez@ipn.mx (Y.K.M.F.); liliana.alamilla@gmail.com (L.A.B.); mmarquezle@ipn.mx (M.M.L.); 2Université Paris-Saclay, INRAE, AgroParisTech, Micalis Institute, Domain de Vilvert, 78350 Jouy-en-Josas, France; luis.bermudez@inrae.fr

**Keywords:** *Lactiplantibacillus plantarum*, synbiotic, microencapsulation, spray drying, dextran sodium sulfate, ulcerative colitis

## Abstract

Background: Microencapsulation improves the storage, handling, and administration of probiotics by protecting them from environmental factors and adverse conditions in the gastrointestinal tract. This process facilitates their controlled delivery in the body, which can simplify their use in therapies without compromising their therapeutic efficacy. Objectives: This study investigates the microencapsulation of *Lactiplantibacillus plantarum* LM-20, its probiotic properties, and its effects in a murine model of ulcerative colitis. Methods/Results: Synbiotic microencapsulation was carried out using spray drying with maltodextrin, gum Arabic, and inulin, achieving an encapsulation efficiency of 90.76%. The resulting microcapsules exhibited remarkable resistance to simulated gastrointestinal conditions in vitro, maintaining a survival rate of 90%. The drying process did not compromise the probiotic characteristics of the bacteria, as they demonstrated enhanced auto-aggregation, hydrophobicity, and phenol tolerance. The therapeutic potential of the microencapsulated synbiotic was evaluated in a murine model of dextran sodium sulfate-induced ulcerative colitis. The results revealed that mice treated with microencapsulated *Lactiplantibacillus plantarum* LM-20 showed an 83.3% reduction in the disease activity index (DAI) compared to the ulcerative colitis control group. Moreover, a significant decrease was observed in pro-inflammatory cytokine levels (IL-1β and TNF-α) and myeloperoxidase activity, with values comparable to those of the healthy control group. Conclusions: These findings suggest that microencapsulated *Lactiplantibacillus plantarum* LM-20 could be a promising candidate for therapeutic applications in the prevention and management of ulcerative colitis.

## 1. Introduction

Ulcerative colitis (UC) is a type of chronic idiopathic inflammatory bowel disease (IBD) affecting the colon. It causes continuous inflammation of the superficial mucosa, extending from the rectum to the proximal colon [1,2]. UC presents a complex multifactorial etiopathogenesis, which remains to be fully elucidated. However, it is known to involve genetic factors, environmental factors (lifestyle, pollution, hygiene, and others), microbial factors related to the microbiota (diet, dysbiosis), and immune aspects [3,4]. The quest to understand the pathogenesis of UC has resulted in clinical and experimental evidence indicating that the process of UC development and progression is primarily characterized by dysregulated immune responses and dysfunctions in the intestinal barrier, leading to continuous and progressive damage [5]. Research has mainly focused on immune cells that produce various inflammatory cytokines that contribute to UC pathology, including macrophages that produce tumor necrosis factor (TNF-α), which is one of the most important cytokines involved in inflammation of the intestinal tract [6], and interleukin 1β (IL-1β), which is a critical inflammatory factor in disease progression [7]. Currently, mesalazine (5-ASA, 5-aminosalicylic acid) treats UC. This molecule is a synthetic drug belonging to the family of non-steroidal anti-inflammatory drugs (NSAIDs) and is used in inflammatory diseases of the gastrointestinal tract (GIT). 5-ASA acts mainly on the intestinal mucosa, inhibiting prostaglandin synthesis, eliminating oxygen free radicals, and reducing intestinal permeability. However, its use can lead to side effects such as abdominal pain, nausea, weight loss, diarrhea, and headaches [8,9]. Among other treatments used for ulcerative colitis, corticosteroids stand out, as they are used for the severe acute phase of the disease [10]. Although they have high anti-inflammatory efficacy, they are not suitable for long-term use due to the high likelihood of patients undergoing colectomy [11]. For this reason, alternative therapies are highly recommended nowadays. 

Probiotics represent an attractive and promising alternative. These are live microorganisms that, when administered in adequate amounts, confer a health benefit on the host [12]. Among them, lactic acid bacteria (LAB), such as lactobacilli, streptococci, enterococci, and bifidobacteria, represent the most studied probiotic genera. Indeed, besides providing unique sensorial characteristics and generating organic acids and bacteriocins in food fermentation processes, these microorganisms also improve intestinal function [13,14]. Benefits include balancing the intestinal microbiota, regulating intestinal transit and the acid–base balance in the colon, and modulating the immune system. In addition, they have been found to possess antimicrobial, antioxidant, and anti-inflammatory properties. Some beneficial effects observed in probiotics derive from their ability to survive in the GIT environment, colonize epithelial surfaces, and impact the immune response [13,14,15,16]. Since probiotics play a key role in intestinal inflammation by modulating the gut microbiota, one potential advantage of probiotics over pharmacological treatments is their ability to reduce chronic inflammation by restoring balance in the gut microbiome. Additionally, probiotics may cause fewer side effects and provide intestinal support during the administration of certain antibiotics [17]. 

In recent years, there has been an increased focus on developing new pharmacological strategies to enhance the viability of probiotic microorganisms. One promising approach is spray-drying microencapsulation, a technique that involves enveloping the probiotic microorganism in a protective matrix that ensures stability and resistance, only altering it when releasing its contents in specific environments. This method helps preserve the microorganism throughout its incorporation into the dosage and storage systems, as well as during its passage through the gastrointestinal tract. In spray drying, a suspension or solution containing probiotics (both the bacteria and the encapsulating matrix) is atomized into fine droplets in a hot air stream, which quickly evaporates the solvent and forms solid microencapsulated particles. This technique provides several benefits, including protection from unfavorable conditions, controlled release, and enhanced stability, making it suitable for use in functional foods, pharmaceuticals, and dietary supplements [18,19]. Probiotics have shown significant potential in treating gastrointestinal diseases such as inflammatory bowel disease, irritable bowel syndrome (IBS), and antibiotic-associated diarrhea by modulating gut microbiota, alleviating inflammation, and restoring intestinal balance. However, their effectiveness can be limited by poor survival in the harsh gastrointestinal environment and reduced stability during storage. Microencapsulation addresses these challenges by shielding probiotics from stomach acid and digestive enzymes, enabling controlled release in the intestines, particularly in the colon, where many gastrointestinal disorders occur. Additionally, microencapsulation enhances the stability of probiotics, extending their shelf life and ensuring their efficacy. Clinical studies have demonstrated that microencapsulated probiotics improve treatment outcomes, including prolonged remission in IBD, symptom relief in IBS, and reduced recurrence of antibiotic-associated diarrhea. As a result, microencapsulation optimizes the therapeutic potential of probiotics and holds significant promise for the effective management of gastrointestinal conditions [20,21,22,23].

Synbiotics are a combination of probiotics and prebiotics that can increase the beneficial effect of probiotics. Microencapsulation is an efficient approach to protecting synbiotics during their passage through the intestinal tract [24]. 

This study aims to evaluate the impact of synbiotic microencapsulation by spray drying on the viability, management, and probiotic properties of the bacterial strain *Lactiplantibacillus plantarum* LM-20, using the Gea-Niro^®^ Mobile Minor 2000 dryer, of which there is little scientific evidence of its use in the microencapsulation of probiotics under the conditions proposed in this work. It also seeks to assess its probiotic potential in a murine model of ulcerative colitis. The choice of the bacterial strain *Lactiplantibacillus plantarum* LM-20, which has probiotic potential, is based on the analysis conducted by Hernández-Delgado et al. (2021) [25], who previously isolated this strain from the fermented solid residue of agave (*Agave angustifolia* Haw) and identified and characterized it. Notably, the strain *Lactiplantibacillus plantarum* LM-20 exhibits high tolerance to acidic conditions, resistance to bile salts, and the ability to form biofilm, which enhances its adhesion to the mucosa and consequently its colonization. Additionally, it demonstrates the ability to modulate the immune response by regulating the production of anti-inflammatory cytokines, which may help reduce inflammation associated with colitis. 

## 2. Materials and Methods

### 2.1. Bacterial Strains

The bacterial strain with probiotic potential (study strain) *Lactiplantibacillus plantarum* LM-20 CDBB-B-2104 (CINVESTAV, Zacatenco, Mexico City, Mexico) and the reference probiotic strain (control strain) *Lactiplantibacillus plantarum* Lp-115 ATCC SD5209 (Danisco, Brabrand, Denmark) [26], whose codes are LM20 and Lp115, respectively, were used. The *Lactiplantibacillus plantarum* LM20 strain belongs to the biological collection of the Food Research Laboratory of the National School of Biological Sciences, Zacatenco Campus of the National Polytechnic Institute; this strain was isolated by Hernández-Delgado et al. (2021) [25] from the fermented solid residue of agave in wooden vats, fermented for 8 to 10 days at room temperature of 25 °C (*Agave angustifolia* Haw), during the production of mezcal in the town of Macuilxóchilt de Ártiagas, in the State of Oaxaca, Mexico.

### 2.2. Growth Conditions of Bacterial Suspension

The strains of *Lactiplantibacillus plantarum* (LM20 and Lp115) were seeded separately on Man, Rogosa, and Sharpe agar (MRS) (Merck^®^, Darmstadt, Germany) and incubated at 37 °C (microbiological incubator model SMI7, Shel-Lab^®^, Cornelius, OR, USA) for 48 h. After incubation, individual isolated colonies were separately activated in 10 mL of sterile Man, Rogosa, and Sharpe broth (MRS) (Merck^®^, Darmstadt, Germany) at 37 °C for 48 h. The bacterial suspension for expansion was obtained by separately inoculating the active strains at 5% (*v*/*v*) in 60 mL of sterile MRS broth under incubation at 37 °C for 48 h and stored in refrigeration at 4 °C. 

### 2.3. Characterization and Bacterial Identification

Colonial morphology was evaluated using the streak plate method, with a bacterial inoculation loop on MRS agar plates incubated at 37 °C for 48 h. The microscopic analysis was performed using Gram staining, observing the morphology and coloration under the optical microscope (VELAB, VE-BC1, Mexico) through the 100× objective; for the catalase test, two drops of 3% hydrogen peroxide (H2O2, Merck, Emsure^®^ ACS, ISO, Darmstadt, Germany) were added to a bacterial colony isolated on a slide [27,28].

The confirmation of the species for the strain isolated by Hernández-Delgado et al. (2021) [25] was carried out using PCR targeting the 16S rRNA gene, using the universal primers 27F (5′-AGAGTTTGATCMTGGCTCAG-3′) and 1492R (5′-TACGGYTACCTTGTTACGACTT-3′). This service was requested and performed at CINVESTAV, Zacatenco, Mexico City, Mexico.

### 2.4. Quantification of Bacterial Concentration

The concentration of viable bacteria (bacterial viability) was determined by the standard plate counting method [29]; serial dilutions (1:10) of the bacterial suspension were performed with phosphate buffer solution (PBS) pH 6.9. The dilutions were seeded in duplicate on plates with MRS agar, incubated at 37 °C for 48 h, and the count of colony-forming units (CFU) was performed; the bacterial concentration was reported in CFU per milliliter (CFU/mL), according to what is mentioned in the Official Mexican Standard NOM-092-SSA1-1994.

### 2.5. Synbiotic Microencapsulation

The bacterial biomass of LM20 and Lp115 (separately) were obtained from the bacterial suspension for expansion generated in the growth conditions of the bacterial suspension. This bacterial suspension was inoculated at 10% (*v*/*v*) in 150 mL of sterile MRS broth and incubated at 37 °C for 48 h. Subsequently, the resulting bacterial suspension was inoculated again at a concentration of 10% (*v*/*v*) in 1200 mL of sterile MRS broth and incubated at 37 °C for 48 h. The cells were collected by centrifugation at 12,560× *g* for 7 min at 4 °C (refrigerated centrifuge Metrix^®^ Dynamic, Velocity 18R, Mexico). The sediment was washed twice with PBS pH 6.9 by centrifugation (12,560× *g* for 7 min at 4 °C) and reconstituted in 30 mL of sterile purified water. The bacterial biomass obtained in this process presented a concentration of 10^12^ CFU/mL viable bacteria. 

A combination of maltodextrin DE10 (Globe^®^ Plus, Ingredion, Westchester, IL, USA) (MD) and gum Arabic (Meyer^®^ Lot: TG0914452, Meyer, St. Louis, MO, USA) (GA) at an individual concentration of 10% *w*/*v* (30 g) was used as wall material; standard Inulin Orafti^®^GR (Azelis Mexico, SA de CV, Tlalnepantla, Mexico) (In) at 5% *w*/*v* (15 g) was used as a prebiotic, and the bacterial suspension of *Lactiplantibacillus plantarum* LM20 or Lp115 was incorporated as a probiotic. The wall material and the prebiotic were dissolved in sterile purified water supported by a magnetic stirrer bar in a hotplate magnetic stirrer (Benchmark Scientific, model 58-H4000-HS, Sayreville, NJ, USA). Maltodextrin (MD) was hydrated 24 h before the drying process. Finally, one hour before starting the drying process, the bacterial suspension was incorporated into the dispersion under constant stirring with the support of a magnetic stirrer bar in a hotplate magnetic stirrer. The final volume of the microencapsulating synbiotic solution was 300 mL, with a percentage of total solids of 25% and a viable bacterial concentration of 10^11^ CFU per gram of total solids. Microencapsulation was performed using the spray drying equipment (Gea-Niro^®^, model Mobile Minor 2000, Soeborg, Denmark) of stainless steel, with a dispersion nozzle 1 mm in diameter, where the microencapsulating synbiotic solution was atomized with the support of a peristaltic pump (Watson Marlow^®^, model 520S, Wilmington, MA, USA) at a dosing speed of 10–15 mL/min (5–8 rpm), with an atomizing pressure of 2 bar, an inlet temperature of 160 ± 2 °C, and an outlet temperature of 80 ± 2 °C. The microencapsulated synbiotic product (microcapsules), obtained after the drying process, was recovered using a sterile spatula and collected in a sterile glass jar. Subsequently, it was stored in a desiccator at room temperature 25 °C. Microencapsulation of the prebiotic was also performed without the bacterium, using the same drying process conditions.

### 2.6. Percentage of Encapsulation Yield

The percentage of encapsulation yield (EY) of the microencapsulated synbiotic obtained was calculated using Equation (1) [30,31]:(1)$$\%EY=(\frac{M}{M_{0}})\times 100$$

M_0_ represents the grams of total solids in the microencapsulating synbiotic solution and M represents the grams of solids (microcapsules) recovered after the spray-drying process.

### 2.7. Moisture Content and Water Activity

Moisture content was determined by placing 0.5 g of the microencapsulated synbiotic on a moisture analyzer (Ohaus^®^, MB45 AM, Mexico City, Mexico) at 100 °C for 10 min. Water activity was determined with the support of a water activity meter (Graigar, HBD5-MS2100Wa, Shenzhen, China) using approximately 0.1 g of the microencapsulate [30]. Both tests were performed in duplicate.

### 2.8. Bacteria Viability and Encapsulation Efficiency

To evaluate bacterial viability, a sample (1 g) of the microencapsulated synbiotic was reconstituted in PBS pH 6.9 (9 mL), with vortex support for 5 min to release the bacteria. Serial dilutions (1:10) were made with PBS pH 6.9 and the dilutions were seeded in duplicate in plates with MRS agar and incubated at 37 ° C for 48 h, while the counting of colony-forming units (CFU) was performed. The concentration was reported in CFU per gram (CFU/g) by the provisions of Official Mexican Standard NOM-092-SSA1-1994 [29]. 

The percentage of encapsulation efficiency (EE) corresponds to the bacterium’s survival rate after the microencapsulation process (viability); this percentage is calculated using Equation (2) [32]: (2)$$\%EE=(\frac{N}{N_{0}})\times 100$$

N_0_ represents the Log_10_ CFU/g of the viable bacteria present in the microencapsulating synbiotic solution before the drying process and N represents the Log_10_ CFU/g of the viable bacteria present in the microcapsules after the spray-drying process.

### 2.9. Morphology and Particle Size

The morphology of the microcapsules obtained was determined by Scanning Electron Microscopy (FEI-ThermoFisher Scientific, ESEM Quanta FEG 250, Waltham, MA, USA). The analysis was performed under a voltage of 15 kV (kilovolts) using an increase of 5000×. The particle size was determined using the ImageJ program (version 1.53t) (National Institutes of Health, USA).

### 2.10. Tolerance to Simulated Gastrointestinal Conditions In Vitro

The methodology used for tolerance to simulate gastrointestinal conditions in vitro was based on what was described by Minekus et al. (2014) [33] with slight modifications. Sterile electrolyte solution A (6.2 g/L Meyer^®^ NaCl, 2.2 g/L KCl J. T. Baker^®^, 0.22 g/L J. T. Baker^®^ CaCl_2_, 1.2 g/L Meyer^®^ NaHCO_3_) and sterile electrolyte solution B (7.5 g/L Meyer^®^ NaCl, 2.4 g/L KCl J. T. Baker^®^, 0.22 g/L J. T. Baker^®^ CaCl_2_, 7.6 g/L NaHCO_3_ Meyer^®^) were prepared, both with distilled water. The simulated salivary fluid was prepared by dissolving 1 g/L lysozyme (Sigma-Aldrich, St. Louis, MO, USA) in sterile electrolyte solution A (adjusted to pH 7.0 ± 0.2 with NaOH), the simulated gastric fluid contained 3 g/L pepsin (Sigma-Aldrich) in sterile electrolyte solution A (adjusted to pH 2.5 ± 0.1 with HCl), and the simulated intestinal fluid was prepared by dissolving 5 g/L bile salts (BioBasic^®^) and 1 g/L pancreatin (Sigma-Aldrich^®^) in a solution of sterile electrolyte B (adjusted) at pH 7.5 ± 0.2 with NaOH). The survival of bacterial strains was determined by sequential exposure of samples of 1 g of synbiotic and 1 mL of bacterial suspension, separately (for each strain), with the same concentration, to simulated gastrointestinal fluids. The sample was placed in 9 mL of simulated salivary fluid for the oral phase for 5 min. Following this, for the gastric phase, 15 mL of simulated gastric fluid was added to the oral phase, which lasted for 120 min. Finally, for the intestinal phase, 80 mL of simulated intestinal fluid was added to the gastric phase, which also lasted for 120 min. In order to simulate the temperature and peristaltic movement of the digestive system, water bath equipment with agitation (PolyScience^®^ Shaking Water Bath System, model 042900, Niles, IL, USA) was used. During digestion, the samples were incubated at 37 °C at 80 rpm. From each simulated phase, an aliquot of 1 mL was taken every 60 min to quantify viable bacteria as described in the Quantification of Bacterial Concentration section.

### 2.11. Probiotic Characterization

A comparison was made between the bacterial strain without microencapsulation, which will be identified as non-microencapsulated, and the bacterial strain present in the microcapsules, which will be identified as microencapsulated, to evaluate the probiotic properties of the bacterial strains before and after the microencapsulation process. 

Phenol tolerance: The phenol tolerance test was performed based on the method described by Singhal et al. (2019) [34] with some modifications: 1 mL of non-microencapsulated bacterial suspension incubated at 37 °C for 24 h and 1 mL of bacterial suspension released from the microcapsules, as described in the *Bacterial Viability and Encapsulation Efficiency* section, were separately transferred to 100 mL of MRS broth with 0.4% phenol (*v*/*v*) and phenol-free MRS broth. Bacterial viability was quantified after 24 h of incubation at 37 °C, using the plate count technique, as detailed in the *Quantification of Bacterial Concentration* section. The test was performed in duplicate. The percentage of phenol tolerance is calculated with Equation (3):(3)$$\%Phenol\;tolerance=(\frac{N_{withphenol}}{N_{withoutphenol}})\times 100$$

N_withphenol_ represents the Log_10_ CFU/mL of the viable bacteria present in MRS broth with phenol and N_withoutphenol_ represents the Log_10_ CFU/mL of the viable bacteria present in phenol-free MRS broth.

Auto-aggregation and hydrophobicity: Auto-aggregation (A) and hydrophobicity (H) tests were determined as previously reported [35,36], with slight modifications. Briefly, for both cases, a non-microencapsulated bacterial suspension, incubated at 37 °C for 24 h, and a bacterial suspension released from the microcapsules, as described in the *Bacteria viability and encapsulation efficiency* section, were used. The cells were separately collected by centrifugation at 12,560× *g* for 7 min at 4 °C (refrigerated centrifuge Metrix^®^ Dynamic, Velocity 18R, Mexico), the sediments were washed twice with PBS pH 6.9, and the suspensions were adjusted to an optical density of 0.25 and 1.0 (A0, H0) at 600 nm. For the auto-aggregation test (A), the adjusted bacterial suspension was stirred in the vortex for 10 min, incubated at 37 °C for 24 h, and absorbance was determined (At). For the hydrophobicity test (H), 1 mL of toluene (J. T. Baker^®^, Phillipsburg, NJ, USA) was added to 5 mL of adjusted bacterial suspension, mixed in the vortex for 2 min, and allowed to stand for one hour at 37 °C. The aqueous phase was then recovered, and its absorbance (Ht) was determined at 600 nm. Both tests were performed in duplicate, and the respective percentages of each test were calculated according to Equations (4) and (5):(4)$$\%A=(\frac{A_{0}-A_{t}}{A_{0}})\times 100$$(5)$$\%H=(\frac{H_{0}-H_{t}}{H_{0}})\times 100$$

### 2.12. Animal Model of Ulcerative Colitis

#### 2.12.1. Animals

Ninety-six female C57BL/6J mice (18–24 g) were used. They were housed individually in acrylic mice cages (27 × 37 × 15 cm in size) under standard conditions of temperature (22–24 °C) and humidity (50–55%) with light/dark cycles of 12 × 12 h. They were fed a standard rodent diet (Rodent Diet 5001 LabDiet^®^, Hubbard, OR, USA) and water ad libitum. After 5 days of acclimatization, the animals were randomized into eight experimental groups (*n* = 12), as detailed in Table 1.

The care and handling of the animals were in agreement with internationally accepted procedures following the recommendations indicated in the Mexican Technical Specifications for the Production, Care, and Use of Laboratory Animals NOM-062-ZOO-1999 [37] (Secretaría de Agricultura, Ganadería, Desarrollo Rural, 2001) and approved by the Institutional Bioethics Committee (CEI-ENCB-ZOO-004-2022; 4 July 2022). 

#### 2.12.2. DSS-Induced Colitis Model

Ulcerative colitis was induced based on the protocols described by Melgar et al. (2005) [38] and Liu et al. (2020) [39], administering, in our case, Dextran Sodium Sulfate grade colitis (DSS; molecular mass: 36,000 to 50,000; MP BiomedicalsTM) at 3% in drinking water from day 8 to day 13. As a pharmacological treatment, mesalazine (5-aminosalicylic acid: 5-ASA) was used at 200 mg/kg. This intestinal anti-inflammatory drug belongs to the salicylate family and is used to treat inflammatory bowel diseases such as ulcerative colitis. The experimental model we propose describes the administration of treatments before the induction of ulcerative colitis, as well as during and after this induction. Therefore, this study can be classified as a therapeutic model focusing on early intervention (prophylaxis), as it seeks to prevent the disease and evaluate its effectiveness in mitigating the clinical manifestations of ulcerative colitis once it has been induced. Based on this, the treatments were administered from day 0 to day 16 (Figure 1) with an oral feeding cannula of curved steel for mice (20-gauge × 1.5-in, 2.25-mm ball tip).

The clinical parameters of induced ulcerative colitis were evaluated by determining the Disease Activity Index (DAI) score [40], which was calculated by combining the score of stool consistency (0 = normal, 1 = slightly loose feces, 2 = loose feces, 2 = loose feces, and 3 = watery diarrhea), bleeding (0 = no bleeding, 1 = slightly bloody, 2 = bloody, and 3= blood in whole colon), and percentage of body weight loss (0 = lower than 1%, 1 = 1–4.99%, 2 = 5–9.99%, 3 = 10% or more), throughout the experimental period. The meaning of the three values corresponds to the DAI value. 

At the end of the experimental period (sacrifice day, Figure 1), the colon was removed, lightly washed with physiological saline solution (PISA^®^) to remove fecal residues, and then weighed and measured to assess changes in weight/length and the presence of adhesions. The presence of adhesions was evaluated following the criteria of Bobin et al. (2020) [41], using a range from 0 to 2, where 0 corresponds to no adhesion presence. Afterward, the collected colon portion was divided into three sections, stored in Eppendorf tubes, and frozen at −70 °C for the subsequent measurement of biochemical parameters and myeloperoxidase catalytic activity.

### 2.13. Colonic Concentrations of TNF-α and IL-1β

The colon samples were weighed and homogenized (1 mg: 5 μL) in protease inhibitor buffer (0.5 mM EDTA, 0.01 mg/mL aprotinin, 0.01 mg/mL pepstatin, 0.01 mg/mL leupeptin, one mM PMSF—C_7_H_7_FO_2_S [0.174 g/ 10 mL DMSO—C_2_H_6_OS], PBS pH 6.9) at 4 °C and centrifuged at 12560xg for 10 min, and the obtained supernatant was collected and used to determine cytokine production with sandwich enzyme-linked immunosorbent assay (ELISA) kits of the sandwich type: TNF-α (FineTest^®^, EM0183, Wuhan, China) and IL-1β (FineTest^®^, EM0109, Wuhan, China ) following the manufacturer’s instructions. 

Protein concentrations in the homogenates were determined using the Bradford colorimetric method, which is based on the absorbance change in Coomassie G-250 when it binds to proteins. To 10 µL of the homogenized sample, 250 µL of Bradford reagent (Thermo-Scientific™) was added and incubated at room temperature for 10 min, after which the absorbance was measured at 595 nm on a Thermo-Scientific Multiskan EX UV-Vis spectrophotometer (ThermoFisher Scientific, Waltham, MA, USA). Protein concentrations were determined by interpolating the absorbance results against a standard curve generated with bovine serum albumin concentrations (0.1–1.0 mg/mL). 

### 2.14. Determination of the Myeloperoxidase Catalytic Activity

The catalytic activity of Myeloperoxidase (MPO) (M6908-5UN, Sigma-Aldrich CO., USA) was determined as reported by Pérez et al. (2019) [42], with slight modifications. MPO was extracted from colon homogenates 1:3 (50 mg of tissue in 150 µL of buffer) by suspension and sonic maceration (Sonics Vibra-Cell VC130 Ultrasonic Processor, USA) of tissue buffered with potassium phosphate 50 mM pH 6, with 0.5% hexadecyltrimethylammonium bromide (HTAB) in cold bath. The homogenized was centrifuged at 13,500 rpm at 4 ° C for 15 min, and an aliquot of 28 μL of the supernatant was taken. This was mixed with 172 μL of potassium phosphate buffer 50 mM pH 6, containing mg/mL of O-dianisidine dihydrochloride and 0.005% H_2_O_2_, and was incubated at room temperature and in darkness for 15 min. MPO activity was determined by absorbance change measured at 492 nm on a Thermo Scientific Multiskan EX UV-Vis spectrophotometer (ThermoFisher Scientific, Waltham, MA, USA). The units (U) of MPO present in the samples were determined by interpolating the absorbance results in the standard curve, which was performed using different units of MPO (0.05–0.80 U MPO) and the same concentration of O-dianisidine dihydrochloride and H_2_O_2_.

### 2.15. Statistical Analysis

The statistical analysis was performed using GraphPad Prism 6 software, expressing the experimental results as mean ± standard error of the mean (SEM) to represent variability. A one-way ANOVA was used to evaluate the significance of differences in the weight–length ratio of the colon, IL-1β, TNF-α, and MPO. A two-way ANOVA was also applied to assess phenol resistance, auto-aggregation, hydrophobicity, tolerance to gastrointestinal conditions, and DAI (the Kruskal–Wallis test was applied to confirm significant differences). Tukey’s post hoc test was used to identify specific differences between groups, with a significance level set at *p ≤* 0.05. Additionally, *p ≤* 0.01, *p ≤* 0.001, and *p ≤* 0.0001 were considered significance levels.

## 3. Results

### 3.1. Bacterial Characterization and Quantification

The characterization report of the bacterial strain *Lactiplantibacillus plantarum* LM20 confirmed that this bacterium corresponds to a Gram-positive *bacterium*, which forms beige colonies, a circular shape, a smooth surface, entire edges, a wet appearance, a milky consistency, and convex elevation in MRS agar (Figure 2) and catalase-negative. According to the BLAST search of the National Center for Biotechnology Information (NCBI) (realized with the support of CINVESTAV, Zacatenco, Mexico City, Mexico), the most similar sequence was the species *Lactiplantibacillus plantarum* strain LLY-606 with the accession number CP023306.1 with a sequence coverage of 100% and an identity of 99%.

### 3.2. Physical Characterization of the Microencapsulated Synbiotics

For each drying process, except for the Syn lot, a bacterial suspension of LM20 or Lp115 (separately) was used at a concentration of 10^12^ CFU/mL; after incorporating it into the synbiotic solution (before the drying process), the bacterial concentration was 10^11^ CFU/g. Four drying processes were carried out with a total volume of 300 mL of synbiotic solution for each process, and the results of the physical characterization of the synbiotic microencapsulation obtained are shown in Table 2. 

### 3.3. Morphology and Particle Size

As shown in Figure 3, the synbiotic microcapsules containing either strain *Lactiplantibacillus plantarum* LM20, Lp115, or no bacteria (Microencapsulated inulin) present an irregular shape (almost spherical) with variable sizes (from 2.8 μm to 7.4 μm), smooth surfaces, no presence of visible fractures, and the presence of concavities.

### 3.4. Tolerance to Gastrointestinal Conditions

Table 3 shows the analysis of bacterial survival of *Lactiplantibacillus plantarum* LM20 and Lp115 strains, both non-microencapsulated and microencapsulated (in this case, the bacteria released from the synbiotic was quantified), after undergoing a four-hour simulation that mimics the gastrointestinal digestion process. It is observed that the non-microencapsulated bacteria of the *Lactiplantibacillus plantarum* LM20 and Lp115 strains show a survival rate of approximately 77% and 76%, respectively, compared to the initial value. In contrast, microencapsulated bacteria have a higher survival rate, reaching more than 90% viability in both strains.

### 3.5. Probiotic Characterization

Regarding phenol resistance, Figure 4 shows that the bacterial strains present a viability percentage higher than 80% in the case of non-microencapsulated bacteria and higher than 90% in the case of microencapsulated bacteria, with no statistically significant difference in bacterial growth before and after the microencapsulation process.

Regarding auto-aggregation and hydrophobicity, Figure 5 shows the properties of hydrophobicity and auto-aggregation of *Lactiplantibacillus plantarum* LM20 and Lp115, where it is observed that *Lactiplantibacillus plantarum* LM20 exhibits a significantly higher percentage in both properties, with 69.6% hydrophobicity and 70.5% auto-aggregation, compared to *Lactiplantibacillus plantarum* Lp115, which shows values of 52.8% and 59.9%, respectively. Notably, both strains show a statistically significant increase of approximately 10% for these properties after the microencapsulation process.

### 3.6. DSS-Induce Colitis Model in Mice

#### 3.6.1. Clinical Evaluation of the Disease

The progression of the disease was assessed using the Disease Activity Index (DAI), which is a mean value resulting from the sum of weight loss, stool consistency, and presence of bleeding in experimental animals during the trial period. In Figure 6, we can observe this DAI value, wherein on the final day of the experiment (sacrifice day), the sham group maintained a DAI value close to zero (0.08). In contrast, the DSS group showed the most significant damage, with an average DAI of 2.4 at the end of the experiment, reflecting a significant difference (*p* ≤ 0.0001) compared to the healthy group, indicating the presence of bleeding, diarrhea, and significant weight loss. The 5-ASA-treated group showed a significant clinical improvement, with an average DAI of 1.3 (*p* ≤ 0.001 vs. DSS), representing a 45.8% reduction in disease severity, although still significantly higher than the healthy group (*p* ≤ 0.0001).

The groups treated with the bacterial strains Lp115 and LM20 without microencapsulation showed average DAIs of 1.0 and 0.9, respectively, indicating a significant reduction in disease progression (*p* ≤ 0.0001 vs. DSS), with reductions of approximately 58.3% and 62.5%. The groups treated with the synbiotics SynLp115 and SynLM20 showed a more pronounced effect, with DAIs of 0.6 and 0.4, respectively, representing reductions of 75% and 83.3% (*p* ≤ 0.0001 vs. DSS) and significant differences compared to the 5-ASA group (*p* ≤ 0.05). These values were closer to those of the healthy group. No significant differences were observed between the free and microencapsulated bacteria, although the latter showed a trend toward a more favorable clinical effect. The group treated with microencapsulated inulin showed a DAI similar to the DSS group, with no significant improvements compared to it, but with significant differences compared to the microencapsulated bacterial groups (*p* ≤ 0.001), highlighting the therapeutic superiority of the probiotic treatments. Regarding adhesions, only the DSS group had a value of 1, while all other groups had values of 0. 

#### 3.6.2. Colon Weight-Length Ratio

Figure 7 represents the weight–length relationship of the colon obtained on the day of sacrifice. It is observed that the groups treated with LM20 and Lp115 without microencapsulation show mean weight–length ratios of 0.042 ± 0.007 g/cm and 0.048 ± 0.009 g/cm, respectively. The groups treated with the synbiotics SynLM20 and SynLp115 show mean weight–length ratios of 0.039 ± 0.011 g/cm and 0.041 ± 0.009 g/cm, respectively. These groups show a significant difference (*p* ≤ 0.0001) compared to the group that only received DSS, which presented a mean ratio of 0.076 ± 0.018 g/cm. On the other hand, the group that received DSS and was treated with 5-ASA showed a mean weight–length ratio of 0.059 ± 0.009 g/cm, representing a significant improvement (*p* ≤ 0.01) compared to the DSS group. The weight–length ratio values observed in the groups that received bacteria are similar to those observed in the sham control group, which presented a mean ratio of 0.035 ± 0.012 g/cm. The significant difference in tissue condition between the group treated with 5-ASA and those that received bacterial treatments was *p* ≤ 0.01. The group that received microencapsulated inulin did not show a significant difference compared to the untreated DSS group or the one that received 5-ASA, with a mean ratio of 0.069 ± 0.016 g/cm. The groups that received the bacteria, both in synbiotic and non-microencapsulated, did not show a significant difference between them.

#### 3.6.3. Interleukin IL-1β Colonic Concentrations

Figure 8 depicts the concentration of IL-1β produced in each study group. In the case of the DSS group without any additional treatment, a mean concentration of 700 ± 70.43 pg/mg was recorded. Conversely, the groups that received DSS and treatment with the synbiotics SynLp115 and SynLM20 exhibited mean concentrations of 93.4 ± 10.66 pg/mg and 86.5 ± 7.95 pg/mg, respectively, indicating a significant difference at *p* ≤ 0.0001 compared to the DSS group. These results are similar to those observed in the sham group, where the concentration was 69.8 ± 13.69 pg/mg. The group treated with 5-ASA showed a mean concentration of 425 ± 50.51 pg/mg, representing a significant difference at *p* ≤ 0.01 compared to the DSS group and *p* ≤ 0.001 compared to the synbiotic groups. As for the non-microencapsulated bacteria (Lp115 and LM20), they exhibited mean concentrations of 311.3 ± 27.38 pg/mg and 299.3 ± 42.51 pg/mg, respectively, showing a significant difference at *p* ≤ 0.0001 compared to the DSS group and *p* ≤ 0.05 compared to their synbiotic counterparts. The group that received microencapsulated inulin did not show a significant difference from the DSS group or the group receiving 5-ASA. The groups receiving the bacteria in synbiotic and non-microencapsulated forms showed no significant differences between strains.

#### 3.6.4. TNF-α Colonic Concentrations

The concentration of TNF-α produced in each study group is shown in Figure 9, where it can be observed that the DSS group presents a mean concentration of 244.3 ± 38.98 pg/mg, while when administered with bacteria in a synbiotic form, namely, SynLp115 and SynLM20, mean concentrations of 41.4 ± 4.91 pg/mg and 39.7 ± 6.59 pg/mg, respectively, can be observed. This evidence shows a significant difference at *p* ≤ 0.0001 vs. the DSS group. These results are similar to those reported by the sham group, showing a concentration of 33.4 ± 6.03 pg/mg. The group treated with 5-ASA showed a mean concentration of 136.3 ± 19.57 pg/mg, representing a significant difference at *p* ≤ 0.01 compared to the DSS group and *p* ≤ 0.05 compared to the synbiotic groups. As for the non-microencapsulated bacteria (Lp115 and LM20), they presented mean concentrations of 98.8 ± 12.55 pg/mg and 91.9 ± 12.47 pg/mg, respectively, showing a significant difference at *p* ≤ 0.0001 compared to the DSS group. The group that received microencapsulated inulin did not show a significant difference compared to the DSS group or the group that received 5-ASA. The groups receiving the bacteria in synbiotic and non-microencapsulated forms did not show significant differences.

#### 3.6.5. Myeloperoxidase Activity

In Figure 10, the concentration of MPO is shown, where it can be observed that the DSS group presents a mean concentration of 28.2 ± 2.25 U/g, while when administered with bacteria in synbiotic form, namely, SynLp115 and SynLM20, mean concentrations of 8.4 ± 0.37 U/g and 7.03 ± 0.54 U/g, respectively, are observed. This indicates a significant difference at *p* ≤ 0.0001 compared to the DSS group. These results are similar to those reported by the sham group, showing a concentration of 5.03 ± 0.36 U/g. The group treated with 5-ASA showed a mean concentration of 16.01 ± 1.34 U/g, representing a significant difference at *p* ≤ 0.0001 compared to the DSS group and *p* ≤ 0.001 compared to the synbiotic groups. As for the non-microencapsulated bacteria (Lp115 and LM20), they presented mean concentrations of 10.5 ± 0.69 U/g and 9.92 ± 0.64 U/g, respectively, showing a significant difference at *p* ≤ 0.0001 compared to the DSS group and *p* ≤ 0.05 compared to the 5-ASA group. The group that received the synbiotic without the bacteria did not show significant differences compared to the DSS group or the group that received the drug 5-ASA. The groups receiving the bacteria, both in synbiotic and non-microencapsulated forms, did not show significant differences. 

## 4. Discussion

Mexico is distinguished by its rich variety of native foods, such as agave, cacao, and corn, which are critical in the production of fermented foods and beverages. The fermentation used in the production of fermented foods and beverages promotes the development of an autochthonous microbiota composed of microorganisms that stably colonize surfaces. The consumption of these fermented foods is directly linked to various health benefits, thanks to the microorganisms involved, especially lactic acid bacteria (LAB), which possess probiotic properties [43,44,45]. In this work, the identity of the bacterial strain *Lactiplantibacillus plantarum* LM-20 was confirmed, which was isolated from the fermented solid residue of agave (*Agave angustifolia* Haw) during the production of mezcal, allowing us to establish that this strain corresponds to the genus and species and presents morphological characteristics, like the reference probiotic strain *Lactiplantibacillus plantarum* LP-115, which in turn corresponds to the characteristics of lactic acid bacteria described in the Bergey Manual of Systematic Bacteriology [46]. 

It is possible that the beneficial effects of probiotic or potentially probiotic strains on host health are not manifested due to the significant decrease in their viability during their storage and passage through the gastrointestinal tract. Therefore, the development of pharmaco-biological techniques and strategies that promote the preservation of bacterial viability during a long period of storage and protection from adverse environmental conditions, without compromising their biological characteristics, is of great importance.

A technique that currently stands out in the preservation of probiotics is microencapsulation by spray drying, which helps achieve high cell densities, improves substrate concentration, and acts as a barrier against the release of trapped cells [47]. The high percentage of microencapsulation efficiency and yield, as well as of the bacterial cells retained within the synbiotic product obtained for each of the strains, can be attributed to the increased efficiency of the heat and mass transfer process that occurs in the spray-drying process when high inlet temperatures are used [48]. The results obtained in this study are similar to, and in some cases, surpass those reported by Arepally et al. (2020) [49], who microencapsulated *Lactobacillus acidophilus* using spray drying with maltodextrin at 20% and gum Arabic in concentrations ranging from 0–10% without any prebiotic. They used inlet temperatures between 130 and 150 °C and an outlet temperature of 55 °C, achieving encapsulation efficiencies between 65 and 89.15%, encapsulation yields between 38.11 and 65.23%, moisture contents between 4.59 and 9.05%, and water activities between 0.33 and 0.52. In another study, Avila-Reyes et al. (2014) [50] microencapsulated *Lacticaseibacillus rhamnosus* using spray drying with inulin as the wall material at concentrations between 10 and 20% (though maltodextrin and gum Arabic were also mentioned as potentially usable, without specifying ranges), with inlet temperatures between 135 and 155 °C and outlet temperatures between 60 and 77 °C. They obtained encapsulation efficiencies above 80%, encapsulation yields between 43 and 54%, moisture contents between 2.6 and 3.9%, and water activities between 0.20 and 0.24. Moisture content and water activity are critical factors in the efficiency of probiotic encapsulation via spray drying as they directly influence the stability and viability of encapsulated bacteria. The optimal moisture content for spray-dried microcapsules is 4–7%, and the water activity should be below 0.6 (Arepally et al., 2020 [49]). These values align with the synbiotic microcapsules obtained in the present study. The results could be attributed to the potent combination of maltodextrin, gum Arabic (1:1 ratio), and inulin. Although inulin was used as a prebiotic in this study, it may also strengthen the encapsulating matrix due to its water-retention properties, helping to decrease water activity and favor bacterial stability. Maltodextrin and gum Arabic, for their part, possess binding and gelling properties that enhance microcapsule formation and improve system stability [32,51,52,53].

One of the most critical factors in the microencapsulation of bacteria with probiotic potential is bacterial viability, which is influenced by inlet temperatures and the formulations of the encapsulating material used. The recommended minimum concentration of probiotic bacteria in the product at the time of consumption is 10^6^ CFU/g [54]. The concentration of viable bacteria obtained in the microencapsulate was 1010 CFU/g of the product, representing a 90% encapsulation efficiency. This renders it an optimal product for health benefits.

The results observed for microcapsules of the synbiotic incorporating LM20 exhibiting an irregular morphology resembling a sphere, with variable dimensions ranging from 2.8 μm to 7.4 μm, smooth surfaces, no evidence of visible fractures, and concavities, indicate effective microencapsulation, which maintains the viability of the bacterial strain. These findings are similar to the results obtained by Barajas-Álvarez et al. (2022) [55], who performed microencapsulation of *Lacticaseibacillus rhamnosus* using a formulation containing maltodextrin, gum Arabic, inulin, and other additional components. They also reported microcapsules with similar morphology and a particle size range ranging from 5 to 13 μm, underscoring that the concavities present in the microcapsules are an inherent feature of the microencapsulation process. 

One of the main reasons for the microencapsulation of probiotics is to facilitate their passage through the gastrointestinal tract, withstanding the acidic environment of the stomach, as well as the lengthy journey through the small intestine, allowing a higher concentration of bacteria to survive for colonization of the colon. Following simulated gastrointestinal digestion for 4 h, the survival capacity of microencapsulated *Lactiplantibacillus plantarum* LM20 is illustrated in Table 3. It can be observed that after 4 h of exposure to the simulated fluid, a release of approximately 90% of bacteria is observed, which is favorable as it promotes the delivery of probiotics to their site of action. This result is consistent with the findings of Kalita et al. (2018) [56], who reported a bacterial release of 75%. Microencapsulation enhances the viability and efficacy of probiotics by acting as a protective barrier in the gastrointestinal tract. This process safeguards bacteria from adverse conditions, such as gastric acidity, minimizing damage to their cell membranes. Additionally, it enables the controlled release of probiotics, optimizing their colonization in the intestine in response to stimuli like pH changes. In summary, microencapsulation optimizes the protection and release of probiotics, improving their effectiveness in treating gastrointestinal diseases [57]. 

Among the probiotic properties is phenol resistance, which is present in the gastrointestinal tract (GIT) and results from the deamination of certain aromatic amino acids (phenylalanine, tyrosine, and tryptophan) derived from the diet. This can affect certain probiotic strains’ bacteriostasis [58,59]. These bacterial strains exhibit a viability percentage exceeding 80% in the case of non-microencapsulated bacteria and surpassing 90% for microencapsulated ones, without a statistically significant difference in bacterial growth before and after the microencapsulation process. This indicates that the deamination of aromatic amino acids by intestinal bacteria does not have a bacteriostatic effect on the analyzed strains. This resistance to phenols in microencapsulated probiotics is mainly due to the protection provided by the microencapsulation process, which creates a physical barrier that limits direct contact between the phenols and the bacterial cell membrane, reducing the damage these substances can cause. Furthermore, by maintaining the stability of the probiotic cells, their structure and functionality are better preserved, allowing them to withstand adverse conditions such as the presence of phenolic compounds, which would otherwise affect their viability.

An important characteristic of strains with probiotic potential is the ability to adhere to the intestinal epithelium, determined by hydrophobicity and auto-aggregation tests, which correspond to the ability to adhere to solvents and auto-aggregation bacteria. Regarding *Lactiplantibacillus plantarum* LM20 and Lp115, we observed that both properties are statistically significantly increased after the microencapsulation process. The hydrophobicity [60] is classified as low if the percentage is between 0 and 29%, medium between 30 and 50%, and high between 51 and 100%, which allows us to classify the strains as bacteria with a high hydrophobicity because *Lactiplantibacillus plantarum* LM20 and Lp115 have hydrophobicity percentages of 80.8 and 61.3%, respectively, after the microencapsulation process. Regarding auto-aggregation, it was observed that after the spray-drying process, the strains showed an increase in the auto-aggregation percentage, exceeding 75%. This result suggests that microencapsulation, by protecting the bacteria from external conditions, preserves the integrity of the cell structure, preventing damage to bacterial membranes both during the encapsulation process and their passage through the gastrointestinal tract. This, in turn, promotes a greater capacity for auto-aggregation [36,61]. 

Regarding the animal model, this study shows that the microencapsulated bacteria in the synbiotic product, specifically the synbiotic with the strain *Lactiplantibacillus plantarum* LM-20, present the least evidence of the development of the disease.

During the experiment, the animals underwent various behavioral changes, including restlessness, hair loss, biting, and other signs of physical and behavioral stress. In the healthy control group, the mice maintained normal behavior, with no significant weight loss or changes in activity or fecal consistency.

In the mice treated with DSS to induce ulcerative colitis, signs of stress, weight loss, and behavioral changes began to appear from day 9, with a significant increase in the DAI (disease activity index), especially from day 12 onward. This pattern of deterioration is similar to that reported by Alex et al. (2009) [62], who used 3% DSS in C57BL/6J mice and observed increased weight loss, diarrhea, and bleeding, confirming the reproducibility of the DSS-induced colitis model.

The DSS + 5-ASA group, which received mesalazine, showed moderate improvement compared to the DSS group, with reduced disease severity, but symptoms persisted, such as watery stools and slight weight loss. These results align with Ren et al. (2023) [63], who also used the C57BL/6J strain with DSS and found reduced inflammation and symptoms without complete disease resolution. While effective in moderate phases of colitis, Mesalazine did not eliminate the disease, as observed by Beiranvand (2021) [8], who highlighted mesalazine’s safety and common side effects, such as abdominal pain and diarrhea.

Regarding the treatments with *Lactiplantibacillus plantarum* strains (Lp115 and LM20), both in free form and microencapsulated, the treated mice remained calm, showing no significant weight loss or stress signs. These findings are consistent with those reported by Daniel et al. (2006) [64], who also used the *Lactiplantibacillus plantarum* Lp115 strain in C57BL/6J mice with TNBS-induced colitis (a different model than DSS) and found improvements in disease symptoms, such as reduced weight loss and normal stool consistency. Furthermore, the results obtained in this study align with the findings of Liu et al. (2020) [39], Wang et al. (2022) [65], and Yu et al. (2023) [66], who also reported that *Lactiplantibacillus plantarum* strains improved clinical parameters in ulcerative colitis models, showing similar therapeutic effects such as reduced weight loss, minimal or no blood presence, regular or soft stool consistency, and significant improvements in DAI values. Microencapsulation of these strains (DSS + SimLp115 and DSS + SimLM20) showed additional benefits, improving the protection and release of bacteria in the intestine, leading to a more significant clinical improvement, approaching the values observed in healthy animals. This finding supports the results of Valcheva et al. (2019) [67], who showed that microencapsulation enhances the therapeutic efficacy of probiotics by allowing more targeted release at the site of action.

The DSS + MI group, which received microcapsules containing only inulin (without bacteria), showed symptoms like the DSS group, suggesting that the prebiotic alone did not have as notable a therapeutic impact as the probiotic bacteria. However, a comparative improvement was observed over the DSS group, suggesting that the prebiotic may modulate the gut microbiota without the intense effects observed with the bacteria, as indicated in previous studies like Valcheva et al. (2019) [67].

Inflammatory diseases, such as ulcerative colitis, are distinguished by chronic inflammatory disorders caused by an abnormal immune response driven by an intense release of cytokines and the infiltration of immune system cells. Various cytokines produced by immune cells are characteristic of UC, most notably tumor necrosis factor-alpha (TNF-α) and interleukin 1β (IL-1β), critical drivers of inflammatory impairment. During the inflammation process, antigen-presenting cells, such as macrophages and dendritic cells, trigger the activation and differentiation of CD4+ T lymphocytes. T lymphocytes differentiate into T helper (Th)-1 and Th-2 cells. Th-1 cells secrete interferon gamma (IFNγ) and interleukin (IL)-2, IL-12, and IL-18, while Th-2 cells secrete IL-4, IL5, IL-6, IL-10, and IL-13. The interferon-gamma of the Th-1 response stimulates macrophages/monocytes to produce TNFα and IL-1B [68,69]. The results obtained in this study show that the bacterial strains used have a positive effect on restoring colonic function and intestinal motility, which is consistent with the findings of Liu et al. (2020) [39], Wang et al. (2022) [65], and Yu et al. (2023) [66], who also reported significant differences in the colon weight–length ratio when using *Lactiplantibacillus plantarum* strains of Chinese origin. In terms of inflammatory modulation, a significant reduction in IL-1β and TNF-α levels was observed in the groups treated with bacteria, compared to the DSS group, supporting the effectiveness of these strains in regulating the intestinal inflammatory response.

These studies also documented a significant decrease in IL-1β and TNF-α production with *Lactiplantibacillus plantarum* treatment, suggesting a potential therapeutic benefit in managing inflammatory bowel diseases. Myeloperoxidase (MPO) is a peroxidase enzyme, which is part of the primary inflammatory markers secreted by intracellular granules of activated neutrophils and monocytes to a lesser extent. This activity is a hallmark of neutrophil infiltration, contributing to tissue damage [70]. Regarding the measurement of inflammatory mediators, there is a low presence of TNF-α and IL-β, as well as for MPO, demonstrating that the administration of probiotics, in this case, specifically the administration of microencapsulated *L. plantarum* LM20 in the synbiotic system (proposed in this work), decreases inflammation in dextran sulfate sodium-induced UC thanks to its multiple mechanisms of action [17,71] of which the ability to colonize the gastrointestinal tract and compete with harmful pathogens stands out. This contributes to the reduction in dangerous bacterial proliferation, which in turn mitigates the exaggerated immune response characteristic of UC. Additionally, probiotics influence the function of the immune system, promoting immune tolerance and decreasing the production of proinflammatory cytokines such as TNF-α and IL-1β, which contributes to the mitigation of chronic inflammation. In this work, a significant reduction in the clinical evidence of UC is observed when administering pharmacological treatment (5-ASA) vs. microencapsulated bacteria (synbiotic), which shows a similar or equal effect to those presented in the sham group. This indicates that in this case, the synbiotic product, in addition to being used as a vehicle for the administration and management of the bacterial strain, also favors the reduction in inflammation and the evolution of murine UC, highlighting the importance of bacteria-based interventions for improving the health and physical condition of the colon in contexts of DSS induced inflammation. Probiotics exert their effects through various mechanisms, including competition for adhesion sites and nutrients, maintaining intestinal microbiota balance, enhancing mucosal barrier function, promoting mucosal immune tolerance, interfering with inflammatory responses, and inhibiting apoptosis in intestinal epithelial cells. Studies indicate that they may be beneficial in treating ulcerative colitis (UC) by modulating intestinal immunity and reducing proinflammatory cytokines such as TNF-α and IL-1β while increasing anti-inflammatory factors like IL-10. Recent research has shown that several probiotics, including the composite VSL#3, which combines eight beneficial bacterial strains, demonstrate promising results in animal models and treating UC in humans [17,71].

## 5. Conclusions

The use of a combination of maltodextrin and gum Arabic as a wall material (coating) for the microencapsulation process of *Lactiplantibacillus plantarum* LM20 and inulin, using the spray-drying technique, favors the retention, viability, management, and stability of the bacterial strain without altering its probiotic properties. The *Lactiplantibacillus plantarum* LM-20 strain microencapsulated in a synbiotic is the best proposal for the prophylactic treatment of murine ulcerative colitis as there is little evidence regarding the development of this disease in murine models. The *Lactiplantibacillus plantarum* LM-20 strain, isolated from the fermented solid residue of agave (*Agave angustifolia* Haw) during mezcal production, shows optimal qualities to be considered a probiotic strain. These findings not only open new possibilities for applying probiotics derived from natural sources in treating inflammatory bowel diseases but may also significantly impact the food and supplement industries, given the growing interest in probiotics sourced from indigenous and sustainable products like agave.

## Figures and Tables

**Figure 1 nutrients-17-00749-f001:**
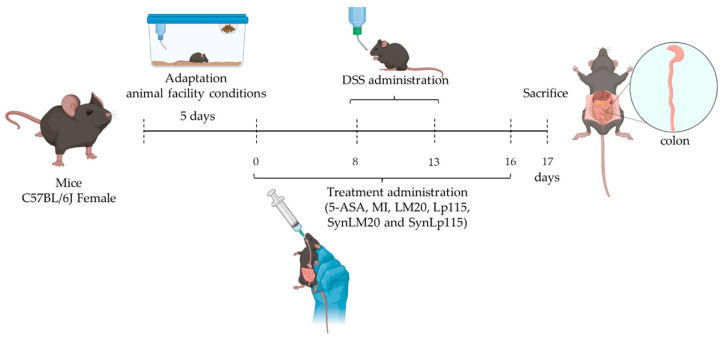
Representation of experiment duration and treatment times. Created with BioRender.com, 2024.

**Figure 2 nutrients-17-00749-f002:**
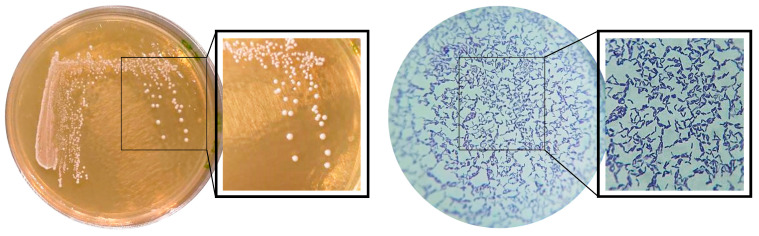
Colony morphology on MRS agar and microscopic morphology of *Lactiplantibacillus plantarum* LM-20 using Gram staining identified under a 100× immersion objective optical microscope.

**Figure 3 nutrients-17-00749-f003:**
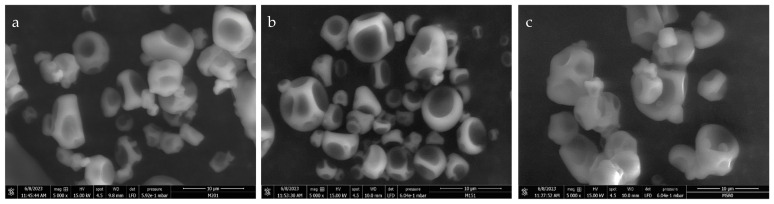
Microscopic morphology of (**a**) SynLM20 and (**b**) SynLp115 synbiotic microcapsules and (**c**) microcapsules without bacteria (Microencapsulated inulin—MI) observed in Scanning Electron Microscopy (FEI-ThermoFisher Scientific, ESEM Quanta FEG 250, USA), under a voltage of 15 kV (kilovolts) and an increase of 5000× (amplification). The particle size was determined using the ImageJ program (version 1.53t) (National Institutes of Health, USA).

**Figure 4 nutrients-17-00749-f004:**
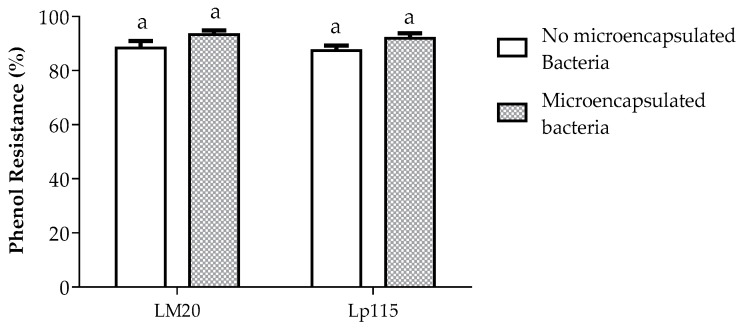
Percentage of resistance to phenol of *Lactiplantibacillus plantarum* LM20 and Lp115, non-microencapsulated and microencapsulated in a synbiotic product by spray drying. Data are represented as mean ± standard error. Bars labeled with the same letters are not significantly different from each other.

**Figure 5 nutrients-17-00749-f005:**
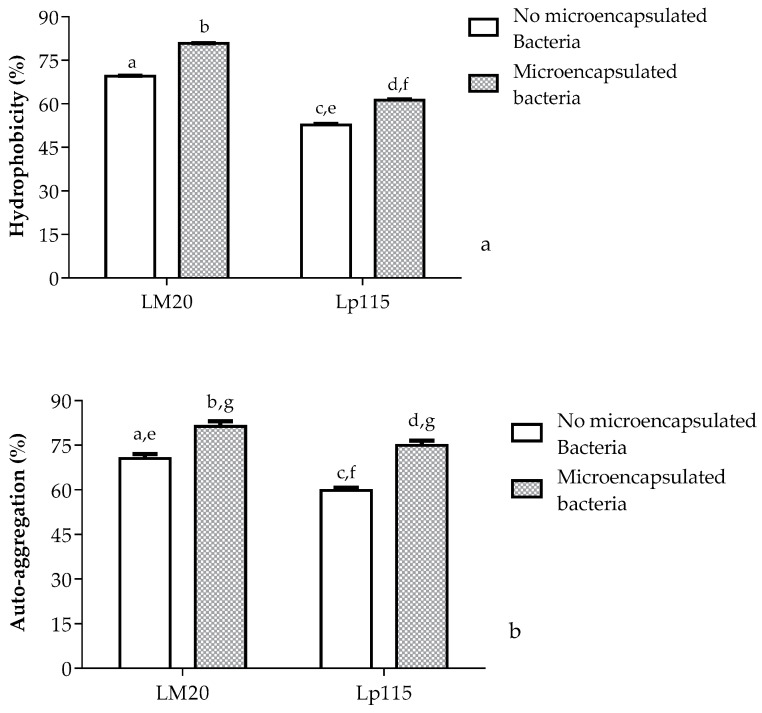
Percentage of hydrophobicity (**a**) and auto-aggregation (**b**) of *Lactiplantibacillus plantarum* LM20 and Lp115, non-microencapsulated and microencapsulated in a synbiotic product by spray drying. Data are represented as mean ± standard error. Bars labeled with the same letters are not significantly different from each other, while those labeled with different letters indicate significant differences from each other. In the case of hydrophobicity, different letters indicate significant differences *p* ≤ 0.0001. For auto-aggregation, the letters “a, b, e, f” indicate significant differences *p* ≤ 0.05, the letters “c, d” indicate significant differences *p* ≤ 0.01, and the letter “g” indicates that there is no significant difference.

**Figure 6 nutrients-17-00749-f006:**
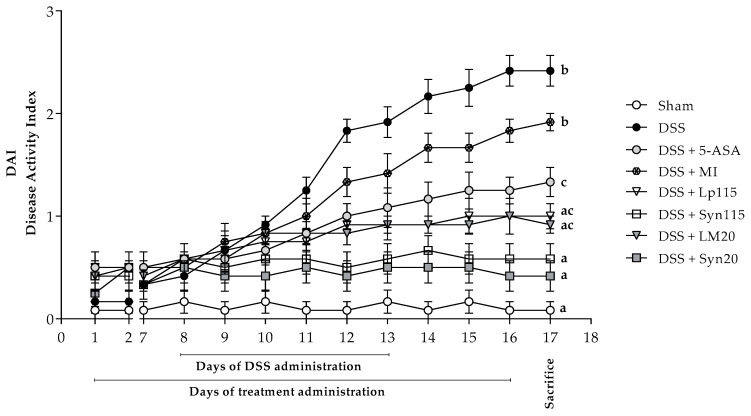
Evaluation of experimental animals’ disease activity index (DAI) during the trial period. From day 1 to day 7 (the day before the administration of DSS), no significant differences were observed, nor during the first 3 days after the administration of DSS. From day 12 onwards, significant differences were observed in the study groups, but for interesting and visual purposes, the results shown correspond to those obtained on the day of sacrifice, *n =* 12. The data are represented as a mean ± standard error. Letters indicate significant differences: bars labeled with the same letters are not significantly different from each other, while those labeled with different letters indicate a significant difference from each other.

**Figure 7 nutrients-17-00749-f007:**
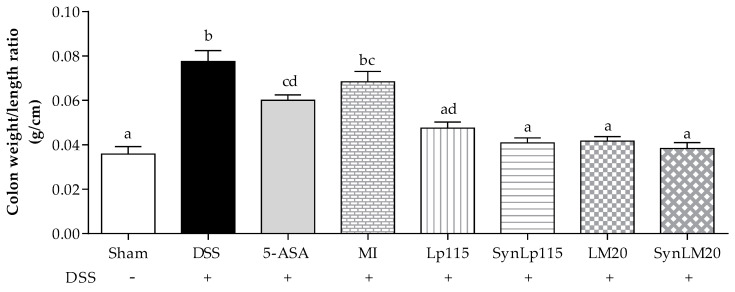
Evaluation of the weight–length ratio of the colon, *n* = 12. Data are represented as mean ± standard error. The letters indicate significant differences: bars labeled with the same letters are not significantly different from each other, while those labeled with different letters indicate significant differences from each other.

**Figure 8 nutrients-17-00749-f008:**
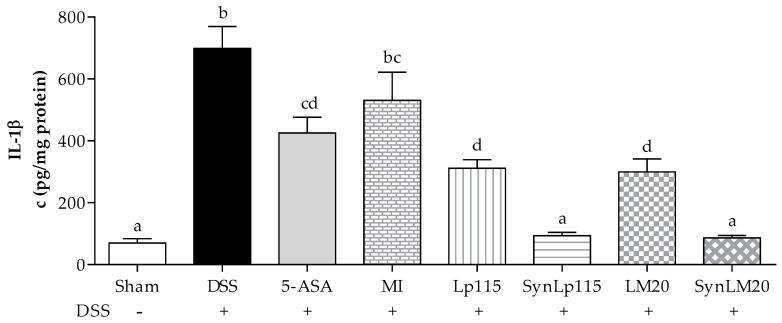
Interleukin IL-1β colonic concentrations of mice with ulcerative colitis induced by DSS, *n =* 12. Data are represented as mean ± standard error. The letters indicate significant differences: bars labeled with the same letters are not significantly different from each other, while those labeled with different letters indicate significant differences from each other.

**Figure 9 nutrients-17-00749-f009:**
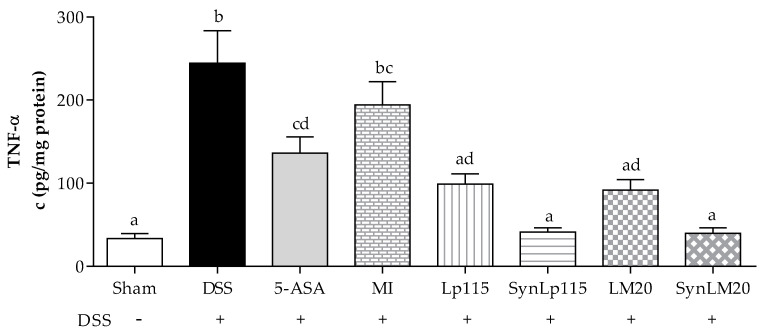
TNF-α colonic concentrations of mice with ulcerative colitis induced by DSS, *n* = 12. Data are represented as mean ± standard error. The letters indicate significant differences: bars labeled with the same letters are not significantly different from each other, while those labeled with different letters indicate significant differences from each other.

**Figure 10 nutrients-17-00749-f010:**
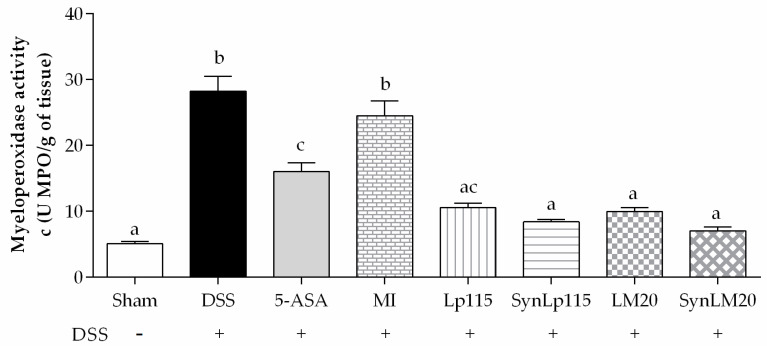
MPO colonic concentrations of mice with ulcerative colitis induced by DSS, *n* = 12. Data are represented as mean ± standard error. The letters indicate significant differences: bars labeled with the same letters are not significantly different from each other, while those labeled with different letters indicate significant differences from each other.

**Table 1 nutrients-17-00749-t001:** Description of experimental groups.

Animals	Group	Description	Administration	Dosage per Day
female mice C57BL/6J *n =* 12 mice per group	Sham	Healthy control	Purified water	300 μL
	DSS	Ulcerative colitis control	Dextran Sodium Sulfate (DSS) grade colitis	3% in purified water (10 mL in the rodent water bottle)
	DSS + 5-ASA	Pharmacological treatment	Mesalazine 5-aminosalicylic acid (5-ASA)	200 mg/kg resuspended in purified water
	DSS + MI	Prebiotic microencapsulated treatment	Microencapsulated prebiotic (Inulin)	0.3 g resuspended in 300 μL purified water
	DSS + LM20	Study bacterial strain treatment	*Lactiplantibacillus plantarum* (LM-20)	bacterial concentration 10^9^ CFU/300 μL in purified water
	DSS + Lp115	Reference probiotic strain treatment	*Lactiplantibacillus plantarum* (Lp-115)	bacterial concentration 10^9^ CFU/300 μL in purified water
	DSS + SynLM20	Synbiotic treatment	Synbiotic microencapsulated *Lactiplantibacillus plantarum* (LM-20)	bacterial concentration 0.3 g/10^9^ CFU/ 300 μL in purified water
	DSS + SynLp115	Synbiotic treatment	Synbiotic microencapsulated *Lactiplantibacillus plantarum* (Lp-115)	bacterial concentration 0.3 g/10^9^ CFU/ 300 μL in purified water

**Table 2 nutrients-17-00749-t002:** Physical characterization of the microencapsulated synbiotics by spray drying.

Synbiotic Formulation			Initial [Bc]	[Bc] in Synbiotic Solution (Before Drying)	[Bc] in Synbiotic Product (After Drying)	Encapsulation Efficiency (%)	Moisture Content (%)	Water Activity (aw)	Encapsulation Yield (%)
Wall Material and Prebiotic	Bacterial Strain	Nomenclature							
Maltodextrin 10%	LM20	SynLM20	12.29 ± 0.10	11.34 ± 0.16	10.29 ± 0.17	90.76 ± 1.58	4.12 ± 0.20	0.23 ± 0.01	88.05 ± 1.91
Gum Arabic 10%	Lp115	SynLp115	12.35 ± 0.16	11.16 ± 0.42	10.01 ± 0.53	89.63 ± 3.00	5.02 ± 0.34	0.27 ± 0.02	84.97 ± 1.85
Standard inulin 5%	-	MI	-	-	-	-	4.06 ± 0.11	0.25 ± 0.01	86.77 ± 1.83

Mean ± standard error, n = 4 drying/microencapsulation processes. [Bc]: bacterial concentration (Log_10_ CFU/mL or g). MI microencapsulated inulin.

**Table 3 nutrients-17-00749-t003:** Counts (Log_10_ CFU/mL or g) of non-microencapsulated or microencapsulated bacteria in simulated gastrointestinal conditions.

Simulation of Gastrointestinal Conditions								
Bacterial Strain	Condition	Initial [Bc]	[Bc] in Simulated Salivary Fluid (5 min)	[Bc] in Simulated Gastric Fluid			[Bc] in Simulated Intestinal Fluid	Bacterial Survival Percentage
				(60 min)	(120 min)	(60 min)	(120 min)	
LM20	Non-microencapsulated	10.43 ± 0.15 10.78 ± 0.42	10.35 ± 0.06 1.87 ± 0.15 ^c^	9.51 ± 0.16 6.59 ± 0.42 ^c^	8.65 ± 0.76 8.09 ± 0.43	8.13 ± 0.44 9.65 ± 0.64 ^a^	8.04 ± 0.13 **** 9.97 ± 0.60 ^b^	77.12%
	Microencapsulated (SynLM20)							92.66%
Lp115	Non-microencapsulated	10.45 ± 0.31 10.55 ± 0.46	10.21 ± 0.26 2.14 ± 0.07 ^c^	8.96 ± 0.39 6.12 ± 0.17 ^c^	8.57 ± 0.48 8.00 ± 0.26 ^a^	8.07 ± 0.59 9.14 ± 0.28 ^a^	8.02 ± 0.80 **** 9.67 ± 0.55 ^b^	76.78%
	Microencapsulated (SynLp115)							91.93%

The simulation of gastrointestinal conditions was performed with three repetitions in duplicate. Data are represented as mean ± standard error. [Bc]: bacterial concentration (Log_10_ CFU/mL or g). The letters (^a^, ^b^*,* and ^c^) indicate significant differences compared to the non-microencapsulated bacterial for each strain, respectively, “^a^” *p ≤* 0.05, “^b^” *p ≤* 0.001, “^c^” *p ≤* 0.0001; **** indicates significant difference compared to the initial concentration vs. final concentration for each non-microencapsulated strain and microencapsulates, respectively, (**** *p ≤* 0.0001).

## Data Availability

All data supporting the conclusions of this study are available in the manuscript. Any questions and/or data can also be requested from the authors if appropriate.

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
