# Peer review of "Microencapsulation and Probiotic Characterization of Lactiplantibacillus plantarum LM-20: Therapeutic Application in a Murine Model of Ulcerative Colitis"

_nutrients, 2025, doi:10.3390/nu17050749_

Round 1
Reviewer 1 Report
Comments and Suggestions for Authors
This excellent article examines the uses of polymer technology (micro-encapsulation) to deliver synbiotics in an animal IBD model. The set-up and work-up are well done. I have some remarks:
Methods: spray drying is used, but matrix polymerization could be better done chemically.
Tables: why is a more common product (e.g. alginate) not used for encapsulation?
Figures: the C57BL/6 mouse model is used, but in a metabolic setting, the BABL/c mouse model could be more appropriate.
Author Response
"Please see the attachment."

For research article
Response to Reviewer 1 Comments
|
||
1. Summary |
|
|
Thank you very much for taking the time to review this manuscript titled “Microencapsulation and probiotic characterization of Lactiplantibacillus plantarum LM-20: Therapeutic application in a murine model of ulcerative colitis” (Journal code: nutrients-3460227). Please find detailed responses below.
|
||
2. Questions for General Evaluation |
Reviewer’s Evaluation |
Response and Revisions |
Does the introduction provide sufficient background and include all relevant references? |
Yes |
|
Is the research design appropriate? |
Can be improved |
|
Are the methods adequately described? |
Yes |
|
Are the results clearly presented? |
Can be improved |
|
Are the conclusions supported by the results? |
Yes |
|
3. Point-by-point response to Comments and Suggestions for Authors |
||
This excellent article examines the uses of polymer technology (micro-encapsulation) to deliver synbiotics in an animal IBD model. The set-up and work-up are well done. I have some remarks:
Comments 1: Response: We appreciate the suggestion of chemical polymerization to form the encapsulating matrix. Although chemical polymerization could offer some advantages, such as better control over the properties of the matrix, it is important to highlight several aspects in which this process could negatively affect the encapsulated probiotic bacteria: · Exposure to extreme conditions: Chemical polymerization, depending on the technique used, may involve exposing the bacteria to high temperatures, reactive chemical agents, or acidic or basic conditions to initiate the reaction. These conditions could damage probiotic bacteria or affect their viability, making it more challenging to ensure they are functional when they reach the intestine. · Formation of cross-links: In some types of chemical polymerization, the formation of cross-links in the encapsulating matrix can reduce solubility or increase the material's rigidity. If the matrix becomes too dense or difficult to dissolve, it could hinder the controlled release of the bacteria at the target site (the intestine), limiting their probiotic effectiveness. · Impact on bioavailability: Excessive rigidity or a non-porous matrix structure could affect the release of the bacteria, preventing the probiotic strains from being adequately released in the intestinal tract, a critical factor for the bacteria to exert their therapeutic effect. · Interactions between polymers and bacteria: Certain chemical compounds used during polymerization could cause undesirable interactions between the polymers and the bacterial cells. This could alter the bacterial surface structure or even trigger bacterial lysis, reducing the number of viable bacteria in the final product.
(Agriopoulou, S., Tarapoulouzi, M., Varzakas, T., & Jafari, S. M. (2023). Application of encapsulation strategies for probiotics: from individual loading to co-encapsulation. Microorganisms, 11(12), 2896.)
For these reasons, we chose to use spray drying, a gentler process that allows the encapsulation of probiotic bacteria without subjecting them to extreme conditions. This process has been proven to be effective in preserving bacterial viability during microencapsulation while also offering adequate control over the size and structure of the encapsulating particles, favoring controlled release in the intestine. · Simplicity and scalability: Spray drying is a continuous and well-established process in the food and pharmaceutical industries, making it easier to implement at a large scale. · Preservation of bacterial viability: Studies have shown that spray drying can maintain the viability of probiotic bacteria during encapsulation. Various studies have evaluated the morphology and viability of different bacterial strains encapsulated using this technique, showing positive results in preserving their biological activity under different storage conditions. · Versatility in the choice of encapsulating materials: Spray drying allows the use of various encapsulating materials, such as alginate, maltodextrin, and chitosan, among others, adapting to the specific needs of the final product. · Production efficiency: This method enables the production of micrometer-sized particles with adjustable distribution, shape, porosity, density, and chemical composition, which is essential for protecting and controlling the release of probiotic bacteria.
(Vivek, K., Mishra, S., Pradhan, R. C., Nagarajan, M., Kumar, P. K., Singh, S. S., ... & Gowda, N. N. (2023). A comprehensive review on microencapsulation of probiotics: technology, carriers and current trends. Applied Food Research, 3(1), 100248.) Nevertheless, we appreciate your suggestion and will consider chemical polymerization for future research, especially if we aim to improve certain specific properties of the matrix.
|
||
|
||
Comments 2:
Tables: why is a more common product (e.g. alginate) not used for encapsulation? Response: We appreciate your comment and understand your concern about using alginate as an encapsulation material. While alginate is a material commonly used for microencapsulation due to its biocompatibility and ability to form gels under specific conditions, in this case, maltodextrin was chosen for several reasons related to the limitations of alginate in the spray drying process. One of the main limitations of alginate is its high viscosity and gelation rate. These characteristics can make it difficult to control and form consistent particles in the spray drying process. Alginate tends to form gels quickly when it comes into contact with water or ions like calcium, which can interfere with the formation of uniformly sized particles. High viscosity can also make it harder to atomize the mixture, potentially leading to an irregular particle size distribution and lower process efficiency. On the other hand, maltodextrin has a lower viscosity, which makes atomization easier and allows for a more uniform particle distribution. Additionally, maltodextrin does not gel as readily, providing better control during atomization. This results in more uniform particles, which are crucial for ensuring controlled release and the viability of encapsulated bacteria. Maltodextrin has also been shown to offer good protective properties for probiotics during drying and storage, ensuring the bacteria maintain viability until release in the intestine. For these reasons, although alginate is an effective material for many applications, maltodextrin was chosen as the encapsulating material to optimize the spray-drying process and ensure the effectiveness and stability of the final product.
Comments 3:
Figures: the C57BL/6 mouse model is used, but in a metabolic setting, the BABL/c mouse model could be more appropriate. Response: We appreciate your suggestion regarding the use of the BABL/c mouse model instead of the C57BL/6 for metabolic studies. However, the C57BL/6 model was specifically chosen for our research for several key reasons that make it particularly suitable for the study of inflammatory bowel diseases (IBD), such as colitis induced by dextran sulfate sodium (DSS). Two inbred mice, specifically the C57BL/6 and BALB/c strains, are most frequently used in chemical induction modeling. Within the same model, different strains show varying sensitivities. C57BL/6 mice are more resistant to 2,4,6-trinitrobenzenesulfonic acid (TNBS) and are widely used for the DSS-induced ulcerative colitis (UC) model, which is primarily characterized by Th1/Th17-mediated immune responses. In contrast, BALB/c mice are more suitable for the TNBS model, which is predominantly mediated by Th1 responses. In this regard, the C57BL/6 model has proven to be robust for studying DSS-induced ulcerative colitis, as it consistently responds to this treatment, showing alterations in the intestinal mucosa and an inflammatory pattern that closely mirrors the pathology of human UC.While the BABL/c model is useful in metabolic studies and diseases such as obesity or diabetes, C57BL/6 is considered one of the best models available for IBD research, as it offers a more accurate representation of the pathophysiology of these diseases, allowing for the evaluation of treatment effects on intestinal inflammation mechanisms. Furthermore, some studies have shown that DSS and the TNBS-induced model better reflect the immunopathological mechanisms of UC and IBD in humans.
(Yang, F., Wang, D., Li, Y., Sang, L., Zhu, J., Wang, J., ... & Sun, X. (2017). Th1/Th2 balance and Th17/Treg‐mediated immunity in relation to murine resistance to dextran sulfate‐induced colitis. Journal of immunology research, 2017(1), 7047201.)
Therefore, the use of the C57BL/6 model in our research aligns with the goal of studying intestinal inflammation, specifically induced by ulcerative colitis with dextran sulfate sodium (DSS). We believe it is the most appropriate model for the specific objectives of our study. In this regard, the symbiotic was administered to evaluate its impact on parameters related to colitis, such as DAI, TNF-α, IL-1β, and myeloperoxidase activity. We appreciate your comment and will consider it for future studies, in which additional metabolic aspects in different models could be explored. |
Reviewer 2 Report
Comments and Suggestions for Authors
I congratulate the Authors for their work
The paper is fluent and enjoyable to read
English language is fine and no writing errors were detected
I Have very few suggestions to improve the paper
line 57-58 Can you find a reference for this statement?
line 366 and following: Please take care of the layout
line 637 and others: You use different forms "gum Arabic" or "Arabic gum". Both forms are accepted, but I suggest to standardise
Author Response
"Please see the attachment."

For research article
Response to Reviewer 2 Comments
|
||
1. Summary |
|
|
Thank you very much for taking the time to review our manuscript titled “Microencapsulation and Probiotic Characterization of Lactiplantibacillus plantarum LM-20: Therapeutic Application in a Murine Model of Ulcerative Colitis” (Article code: nutrients-3460227). Below, you will find our detailed responses. All modifications in the manuscript (nutrients-3460227-Highlighting revisions made) have been made and are highlighted in yellow. We hope that the revised version of the manuscript is now suitable for publication in your esteemed journal.
|
||
2. Questions for General Evaluation |
Reviewer’s Evaluation |
Response and Revisions |
Does the introduction provide sufficient background and include all relevant references? |
Yes |
|
Is the research design appropriate? |
Yes |
|
Are the methods adequately described? |
Yes |
|
Are the results clearly presented? |
Yes |
|
Are the conclusions supported by the results? |
Yes |
|
3. Point-by-point response to Comments and Suggestions for Authors |
||
I congratulate the Authors for their work English language is fine and no writing errors were detected I Have very few suggestions to improve the paper
Comments 1: |
||
Response: Thank you for your comment. We have added the requested reference to support the statement in lines 57-58. Additionally, we have corrected the format of reference 10 (now reference 11), as we noticed that the period was incorrectly placed outside the brackets, and we have adjusted it to comply with the proper format. It is important to mention that the latest version of the manuscript, which the platform requests to be downloaded again due to possible modifications, presents errors in the numbering of the references. Therefore, these have been reviewed and updated in the new file: nutrients-3460227-Highlighting revisions made.
|
||
Comments 2:
line 366 and following: Please take care of the layout Response: Thank you for your comment. We have reviewed and corrected the formatting in line 366 and the following to ensure a proper layout.
Comments 3:
line 637 and others: You use different forms "gum Arabic" or "Arabic gum". Both forms are accepted, but I suggest to standardize. |
Reviewer 3 Report
Comments and Suggestions for Authors
Garfias Noguez and colleagues aimed to demonstrate that microencapsulation of Lactiplantibacillus plantarum LM-20 did not affect the probiotic characteristics of the bacteria. Additionally, the authors used the microencapsulated probiotic in a murine model of ulcerative colitis (DSS) showing that the animal group receiving microencapsulated Lactiplantibacillus plantarum LM-20 exhibited a significant reduction in the disease activity index compared to the DSS control group and to the group receiving the non-microencapsulated probiotic.
Whie the data presented might help to find alternatives to the current treatment for ulcerative colitis, there are few point that need to be further clarified/investigated.
Fig. 2: It should mention in the text. Additionally the figure is wrongly formatted in the manuscript
Fig 4-5: These figures might even go together. Additionally, figure 5A has no error bars, can the authors explain this?
Fig. 6: Why the mice used were housed individually? This might affect the whole experiment since the individually housed animals can experience stress and discomfort that is not related with the treatment.
Fig.8-10: Thee figures can go together. Why cytokines levels were not measured in the serum? To strengthen the effect of the encapsulated probiotic in the DSS model, authors might investigate mRNA expression of tight junction proteins, mucins, inflammatory cytokines (not only IL1-β and TNF-α but also IL-17, Il-4, Il-6 and IFN-ɣ) and immune cell markers (i.e. CD4, CD8, F4/80 and Foxp3).
ARRIVE GUIDELINES: Arrive lines listed by the authors do not correspond with the text.
Minor Comments:
Line 40: UC abbreviation was already mentioned in the manuscript
Lines 9-43: Can the authors rephrase it?
Lines 62-65: Please add references
Lines 66-67: Please add references
Lines 69-74: The sentence is not clear (i.e. “potential limitation are reducing chronic inflammation?”)
Lines 75-84: References are missing
Line 88: IBD abbreviation was already mentioned in the manuscript
Lines 75-100: Some paragraphs are repetitive, please rephrase this part.
Line 297: Which modifications?
Author Response
"Please see the attachment."

For research article
Response to Reviewer 3 Comments
|
||
1. Summary |
|
|
Thank you very much for taking the time to review our manuscript titled “Microencapsulation and Probiotic Characterization of Lactiplantibacillus plantarum LM-20: Therapeutic Application in a Murine Model of Ulcerative Colitis” (Article code: nutrients-3460227). Below, you will find our detailed responses. All modifications in the manuscript (nutrients-3460227-Highlighting revisions made) have been made and are highlighted in blue. We hope that the revised version of the manuscript is now suitable for publication in your esteemed journal.
|
||
2. Questions for General Evaluation |
Reviewer’s Evaluation |
Response and Revisions |
Does the introduction provide sufficient background and include all relevant references? |
Can be improved |
|
Is the research design appropriate? |
Can be improved |
|
Are the methods adequately described? |
Yes |
|
Are the results clearly presented? |
Yes |
|
Are the conclusions supported by the results? |
Can be improved |
|
3. Point-by-point response to Comments and Suggestions for Authors |
||
Garfias Noguez and colleagues aimed to demonstrate that microencapsulation of Lactiplantibacillus plantarum LM-20 did not affect the probiotic characteristics of the bacteria. Additionally, the authors used the microencapsulated probiotic in a murine model of ulcerative colitis (DSS) showing that the animal group receiving microencapsulated Lactiplantibacillus plantarum LM-20 exhibited a significant reduction in the disease activity index compared to the DSS control group and to the group receiving the non-microencapsulated probiotic.
Whie the data presented might help to find alternatives to the current treatment for ulcerative colitis, there are few point that need to be further clarified/investigated.
It is important to mention that the latest version of the manuscript, which the platform requests to be downloaded again due to possible modifications, presents errors in the numbering of the references. Therefore, these have been reviewed and updated in the file: nutrients-3460227-Highlighting revisions made.
Comments 1: |
||
|
||
Comments 2: Fig 4-5: These figures might even go together. Additionally, figure 5A has no error bars, can the authors explain this? Response: Thank you for your comment. Regarding Figures 4 and 5, we decided to keep them separate because Figure 4 shows the results for phenol resistance. In contrast, Figure 5 is divided into 5A, which corresponds to hydrophobicity, and 5B, which shows autoaggregation. Although both figures are related to the analysis of bacterial properties, we wanted to present them separately for better clarity and understanding of the results. Regarding Figure 5A, the error bars were initially tiny and not visible. To address this, we have increased the size of Figure 5 and adjusted the error bars by making them larger and changing their color to black, making them more visible and easier to identify. We hope these adjustments improve the presentation and interpretation of the results.
Comments 3: Fig. 6: Why the mice used were housed individually? This might affect the whole experiment since the individually housed animals can experience stress and discomfort that is not related with the treatment. Response: We agree with your comment. However, in this case, we decided to house the mice individually to precisely control their food and water intake and monitor the signs considered in the disease activity index (DAI): body weight and consistency and fecal blood presence. This also reduces the variability in DSS consumption. In another way, some mice can show signs of discomfort related to colitis induction, and we have observed an increase in aggressiveness in these animals, so we preferred to maintain separation.
Comments 4: Fig.8-10: Thee figures can go together. Why cytokines levels were not measured in the serum? To strengthen the effect of the encapsulated probiotic in the DSS model, authors might investigate mRNA expression of tight junction proteins, mucins, inflammatory cytokines (not only IL1-β and TNF-α but also IL-17, Il-4, Il-6 and IFN-ɣ) and immune cell markers (i.e. CD4, CD8, F4/80 and Foxp3).
Response: Thanks for your observation. With respect, we selected the use of colon tissue rather than serum for cytokine identification in this study because the main goal was to assess the local changes in the colon environment, which is the direct site of inflammation in the ulcerative colitis model. Proinflammatory cytokines are produced at the injury site. They are more accurately reflected in the local immune response than serum, which may not adequately capture the variations in intestinal inflammatory response. Additionally, measurements in colon tissue allow for a more specific analysis of proteins and cytokines associated with intestinal barriers, providing a better understanding of the encapsulated probiotic's effect on local inflammation modulation. We also appreciate the suggestion to investigate mRNA expression of tight junction proteins, mucins, inflammatory cytokines, and immune cell markers. However, this study did not aim to address these aspects as it focused primarily on the characterization of the inflammatory response in terms of classical cytokines (IL-1β and TNF-α) in the colitis model. The mRNA analysis of additional proteins and other immune markers will be considered for future studies by the lead author, who plans further to investigate these aspects as part of a postdoctoral fellowship Finally, Figures 8-10 were presented separately to analyze the results more fluidly and comprehensibly. Keeping each graph separate makes it easier to interpret each parameter evaluated, which helps highlight the differences and relationships between variables. This organization also prevents overloading the figures with too much information, ensuring the data is more precise and easier to follow and analyze individually. We hope you understand our point of view.
Comments 5:
Comments 6:
Comments 7: Lines 9-43: Can the authors rephrase it? Response: Thank you for your suggestion. We have rephrased lines 9–43 to improve clarity and readability while maintaining the original meaning. The revised section can be found in the updated manuscript.
Comments 8: Response: Thank you for your comment. The corrections have already been made in the text, and the references corresponding to lines 66-67 are now included as references 13 to 16, which can be found in line 72 of the clean version of the manuscript uploaded to the platform.
Comments 10: Lines 69-74: The sentence is not clear (i.e. “potential limitation are reducing chronic inflammation?”) Response: We have rephrased the previous paragraph as follows to improve clarity and avoid confusion in the wording.
Comments 11: Lines 75-84: References are missing Line 88: IBD abbreviation was already mentioned in the manuscript Response: Thank you for your comment. We have checked, and the abbreviation IBD was already mentioned earlier in the manuscript. We have adjusted the wording to avoid redundancy.
Comments 13: Lines 75-100: Some paragraphs are repetitive, please rephrase this part. Response: The paragraph has been rephrased to avoid repetition and improve clarity. The revised section can be found in the updated manuscript.
Comments 14: Line 297: Which modifications? Response: According to your comments, we have rewritten all section 2.12, and we have indicated in the new section 2.12.2 the specific modifications in our method. The revised section can be found in the updated manuscript. |
Round 2
Reviewer 3 Report
Comments and Suggestions for Authors
I think that the authors have adequately addressed the comments made by this reviewer in the revised version of the manuscript. However, in the ARRIVE checklist some of the line numbers do not match with the recommendation content. Can the authors please revise it?